# Look at Tempered Subdiffusion in a Conjugate Map: Desire for the Confinement

**DOI:** 10.3390/e22111317

**Published:** 2020-11-18

**Authors:** Aleksander Stanislavsky, Aleksander Weron

**Affiliations:** Faculty of Pure and Applied Mathematics, Hugo Steinhaus Center, Wrocław University of Science and Technology, Wyb. Wyspiańskiego 27, 50-370 Wroclaw, Poland; aleksander.weron@pwr.edu.pl

**Keywords:** anomalous diffusion, statistical analysis, single-particle tracking, trajectory classification

## Abstract

The Laplace distribution of random processes was observed in numerous situations that include glasses, colloidal suspensions, live cells, and firm growth. Its origin is not so trivial as in the case of Gaussian distribution, supported by the central limit theorem. Sums of Laplace distributed random variables are not Laplace distributed. We discovered a new mechanism leading to the Laplace distribution of observable values. This mechanism changes the contribution ratio between a jump and a continuous parts of random processes. Our concept uses properties of Bernstein functions and subordinators connected with them.

## 1. Introduction

Based on a myriad of examples in the physical sciences, 1963 Nobel Prize winner in physics H.P. Wigner emphasized the exceptional role of mathematics in understanding the physical structure of the world around us [1]. Indeed, mathematics is a kind of mental tool created for this purpose, and the world is organized in a logical pattern very similar to mathematics [2]. Thus, mathematics turns out to be the language of science and technology. In many experiments the single-molecule motion manifests anomalous diffusion, absolutely not like the classical Brownian diffusion having the mean-squared displacement (MSD) linear in time [3]. To describe the data, a number of theoretical models was developed. The most popular of them are: continuous-time random walk (CTRW) and Fractional Fokker–Planck equation (FFPE) [4,5,6,7], fractional Klein–Kramers equation [8], obstructed diffusion (OD) [9,10], random walk on random walk (RWRW) [11,12], fractional Brownian motion (FBM) [13,14,15,16], fractional Lévy α-stable motion (FLSM) [17,18,19], fractional Langevin equation (FLE) [20,21] and autoregressive fractionally integrated moving average (ARFIMA), see [22] and references therein. The ARFIMA model [23,24,25] is a discrete time analogue of the overdamped fractional Langevin equation [26] responsible for the non-Gaussian law (Lévy α-stable) and a long memory. Moreover, the ARFIMA process is a universal and simple discrete time model for fractional dynamics of empirical data. Recall also that the celebrated FBM and FLSM is nothing but the limiting case of ARFIMA. Since the ARFIMA models were successful in analyzing data from other fields (econometrics, see 2003 Nobel Prize in Economic Sciences for C.W.J. Granger and R. Engel; finance and engineering [27,28,29]), many statistical tools (and computer packages, e.g., ITMS [24]) are widely available for users, see [25,30].

A relation beetwen physical environment and mathematical models is crucial [1,30]:–Trapping, crowded environment (CTRW, FFPE, subordinated BM);–Labyrinthine environment (OD, percolation, RWRW);–Viscoelastic system (FBM, FLSM, FLE, ARFIMA);–System with time-dependent diffusion (scaled BM, scaled FBM, ARFIMA);–System with transient diffusion (BM with transient subordinators).

By using the conjugated Bernstein function theory [31] for a subordinated diffusion, we uncover here a general universal behavior for the pairs of conjugated subordinators. Namely, one can connect the tempered subdiffusion with the diffusion-limited aggregation. Moreover, for the pure Brownian motion this is the Laplace distribution whereas for the Lévy flights is its generalization, the Linnik distribution. It should be also noticed that a large part of well-known anomalous diffusion processes can be represented as time-changed Brownian motion. Thus, by employing the I. Monroe result [32] we find that an anomalous diffusion process is represented as time-changed Brownian motion if and only if it is a semimartingale, [33]. Randomizing the time of the Brownian motion B(t) by using the independent random process U(t), we obtain a new process X(t)=B(U(t)). Such an operation is called subordination, first introduced by S. Bochner [34], and see also [35]. The process *B*, called the parent process, is directed by the new operational time clock *U* called subordinator. The first reasonable usage of subordination in physics dates back to [36], see also [37,38]. Later, physicists developed a considerable intuition on subordination [39,40]. For the very recent results, see [41].

In the method of single-particle tracking (SPT) a major result is that motion in cell membranes is not limited to pure diffusion. Several modes of motion have been detected including such as immobile, confined, tethered, directed, normal diffusion, and anomalous diffusion [42]. After an ensemble average, the time dependence of the MSD for pure modes of motion is well recognized from others [43,44]. One of important phenomena related to the classification of modes of motion is that practically all experimental results show apparent transitions among modes of motion [45]. If a transition is real, it causes this nonclassical behavior to be different. Their studies are of great interest [46,47,48]. In cell membranes, anomalous diffusion is most likely the result of both obstacles to diffusion and traps with a distribution of binding energies or escape times [49]. Confined motion may result from corrals formed by cytoskeletal proteins near the membrane, from tethering to immobile species, or from restrictions to motion imposed by lipid domains [50]. The confined diffusion of plasma membrane proteins or lipids can be regarded as a special case of subdiffusion [51]. Analytical treatments have been provided for certain shapes of the confinement zones and the characteristic mobilities [52]. The motion of single biomolecules inside a living cell often exhibits subdiffusion in the confined and crowded environment [53]. For the interpretation of experimental results and quantitative predictions of the diffusion behavior, theoretical models can be extremely helpful. One of them is presented in this paper. It describes the transition from subdiffusion to a confined state. It is interesting that the model has a one-to-one connection with the well-known tempered subdiffusion which demonstrates the transition from subdiffusion to normal diffusion in condensed matter physics [54] and geophysics [55], respectively. Such characteristic crossover from subdiffusion to normal diffusion has been observed also in lipid bilayer systems [56,57,58]. The tempered stable process in different guises has been intensively researched recently [59,60,61,62,63,64,65,66,67]. The aim of this work is to get the stochastic representation of anomalous diffusion tending to the confinement. We compare its properties with similar ones for the tempered subdiffusion. Finally, our results are applied for relaxation processes with non-exponential decay as well as for the analysis of the experimental data with confined random trajectories of G proteins and receptors in living cells.

## 2. Conjugate Laplace Exponents and Stochastic Representation of Anomalous Diffusion

The subdiffusive dynamics is fruitfully modeled as a diffusive motion X(τ) subordinated by a wide class of random processes subject to infinitely divisible distributions. If the stochastic process X(τ) has the probability density function (PDF) h(x,τ), it is a solution of the ordinary Fokker–Planck (FP) equation
(1)$$\partial h\left(x,\tau \right)/\partial \tau =\hat{L}\left(x\right)\;h\left(x,\tau \right)\;.$$
where $\hat{L}\left(x\right)$ is the time-independent FP operator (for example, $-\frac{\partial }{\partial x}F\left(x\right)+D\frac{\partial^{2}}{\partial x^{2}}$ with a force *F*). Generally, the operator $\hat{L}$ can be both multidimensional and fractional in space, but with no loss of generality we will consider the one-dimensional case. Infinitely divisible distributions, following the Lévy–Khintchine formula [68], are characterized by the exponentially weighted function
(2)$$\langle e^{-u\;T_{\Psi }\left(\tau \right)}\rangle =e^{-\tau \Psi (u)}=\int_{0}^{\infty }e^{-ut}\;g_{\Psi }\left(t,\tau \right)\;dt\;,$$
where Ψ(u) is called the Laplace exponent, and $g_{\Psi }\left(t,\tau \right)$ is the PDF of this process. Note, the Laplace exponents may be only Bernstein functions. This is a very extensive class of functions [31]. In the theory of Bernstein functions a special role is played by so-called conjugate pairs [69]. If one of them is Ψ(s), then another will take the form Φ(s)=s/Ψ(s). The parent process X(τ) may be subordinated by $S_{\Psi }\left(t\right)=\inf\left\{\tau >0\;:\;T_{\Psi }\left(\tau \right)>t\right\}$ as well as $S_{\Phi }\left(t\right)=\inf\left\{\tau >0\;:\;T_{\Phi }\left(\tau \right)>t\right\}$, where $T_{\Phi }$ is a conjugate subordinator. Both cases lead to the FP equation in a general form
(3)$$p\left(x,t\right)=q\left(x\right)+\int_{0}^{t}d\tau \;M\left(t-\tau \right)\;\hat{L}\left(x\right)\;p\left(x,\tau \right)\;,$$
where M(t) is the memory function [54,70]. The kernel M(t) has a simple expression after the Laplace transform. Denote the inverse Laplace transform $L_{t}^{-1}$. This gives
(4)$$\begin{array}{ccc} M_{\Psi }\left(t\right) & = & \frac{1}{2\pi i}\int_{c-i\infty }^{c+i\infty }e^{st}\;\frac{ds}{\Psi (s)}=L_{t}^{-1}\frac{1}{\Psi (s)}\;, \end{array}$$
(5)$$\begin{array}{ccc} M_{\Phi }\left(t\right) & = & \frac{1}{2\pi i}\int_{c-i\infty }^{c+i\infty }e^{st}\;\Psi \left(s\right)\;\frac{ds}{s}=L_{t}^{-1}\frac{\Psi (s)}{s}\;, \end{array}$$
where *c* is large enough that 1/Ψ(s) (for the first case) and Ψ(s)/s (for the second) are defined for ℜs≥c, and $i^{2}=-1$. The PDF of the operational time $S_{\Phi }\left(t\right)$ is simply written as a Laplace image
(6)$$\tilde{f}\left(\tau ,s\right)=\frac{1}{\Psi (s)}\;e^{-\tau s/\Psi (s)}\;.$$
The solution (propagator) of Equation (Equation 3) takes the form of a subordination integral
(7)$$p\left(x,t\right)=\int_{0}^{\infty }h\left(x,\tau \right)\;f\left(\tau ,t\right)\;d\tau \;.$$
Using the Brownian motion as a parent process
(8)$$h_{B}\left(x,\tau \right)=\frac{1}{\sqrt{2\pi D\tau }}\;\exp\left(-\frac{x^{2}}{2D\tau }\right)\;,$$
the Laplace image $\tilde{p}\left(x,s\right)$ is written as the tabulated integral [71], expressed in terms of the modified Bessel function of the third kind. As its index is equal to 1/2, we get the following propagator
(9)$$\tilde{p}\left(x,s\right)=\frac{1}{\sqrt{2D}}\;\frac{1}{\sqrt{s\Psi (s)}}\;\exp\left(-2\frac{\left|x\right|\sqrt{s}}{\sqrt{2D\Psi (s)}}\right)\;.$$
Similar calculations can be fulfilled for $S_{\Psi }\left(t\right)$, which we will not present here. If the moments of a parent process X(τ) are known exactly, as in the case of the Brownian motion, the moments of the process $X[S_{\Phi }\left(t\right)]$ can be found analytically. Using the MSD of Brownian motion in the form $\langle B^{2}\left(\tau \right)\rangle =D\tau$, where *D* is a diffusive constant, the MSD of $Y\left(t\right)=B[S_{\Psi }\left(t\right)]$ and $Y\left(t\right)=B[S_{\Phi }\left(t\right)]$ reads
(10)$$\begin{array}{ccc} \langle B^{2}\left[S_{\Psi }\left(t\right)\right]\rangle & = & D\;L_{t}^{-1}\frac{1}{s\Psi (s)}=D\int_{0}^{t}M_{\Psi }\left(y\right)\;dy\;, \end{array}$$
(11)$$\begin{array}{ccc} \langle B^{2}\left[S_{\Phi }\left(t\right)\right]\rangle & = & D\;L_{t}^{-1}\frac{\Psi (s)}{s^{2}}=D\int_{0}^{t}M_{\Phi }\left(y\right)\;dy\;. \end{array}$$
It is clear that the MSD is depended on the function Ψ(s), but in different ways it manifests in a conjugate pair. Similar analysis of two different forms of the Fokker–Planck equation where the memory kernels are conjugate pairs has been done in [72,73]. When the memory kernel is an exponentially truncated power-law, the MSD can approach to saturation. In the next section, we will look at specific examples.

## 3. Tempered α-Stable Process and Its Conjugate Partner

An important exemplar of infinitely divisible subordinators is tempered α-stable processes, having all moments of operational time [74]. In this case the diffusive motion demonstrates an intermediate behavior between subdiffusion and normal diffusion [54,55]. Then the Laplace exponent is $\Psi_{\mathrm{temp}}\left(s\right)={\left(s+\delta \right)}^{\alpha }-\delta^{\alpha }$, where δ is a positive constant and 0<α<1. If δ equals to zero, the tempered α-stable process becomes ordinary α-stable. Let the Brownian motion be a parent process, and the inverse tempered α-stable process is directing. The MSD of the subordinated diffusion is
(12)$$\langle x^{2}\left(t\right)\rangle =D\int_{0}^{t}e^{-\delta y}\;y^{\alpha -1}\;E_{\alpha ,\;\alpha }\left(\delta^{\alpha }y^{\alpha }\right)\;dy\;,$$
where $E_{\alpha ,\beta }\left(x\right)=\sum_{k=0}^{\infty }x^{k}/\Gamma \left(\alpha k+\beta \right)$ is the two-parameter Mittag–Lefeffler function [75]. If t≪1 (or δ→0), this value strives for $Dt^{\alpha }/\Gamma \left(\alpha +1\right)$, whereas for t≫1 (or α→1) it is linear in time $D\delta^{1-\alpha }t/\alpha$ as expected for normal diffusion shown in Figure 1a. From the asymptotic values for $\langle x^{2}\left(t\right)\rangle$ it is easy enough to obtain the crossover time $t_{x}={\left[\alpha \;\delta^{\alpha -1}/\Gamma \left(\alpha +1\right)\right]}^{1/(1-\alpha )}$ between the two diffusive modes, also shown in Figure 1a. This diffusion behaves anomalous at short time and almost normal at long times.

Next, we study the diffusion motion with the conjugate Laplace exponent $s/\Psi_{\mathrm{temp}}\left(s\right)$. Its MSD is not difficult to find. It is expressed in terms of the three-parameter Mittag–Leffler function [75], having the following Taylor series
(13)$$E_{\alpha ,\beta }^{\rho }\left(x\right)=\sum_{k=0}^{\infty }\frac{\left(\rho ,k\right)\;x^{k}}{\Gamma (\alpha k+\beta )k!}\;,\;\alpha ,\beta >0\;,$$
where (ρ,k)=ρ(ρ+1)(ρ+2)⋯(ρ+k−1) is the Appell’s symbol with (ρ,0)=1, ρ≠0. The MSD has also an analytical form
(14)$$\begin{array}{ccc} \langle x^{2}\left(t\right)\rangle & = & D\int_{0}^{t}e^{-\delta y}\;y^{-\alpha }\;E_{1,\;1-\alpha }\left(\delta y\right)\;dy-D\delta^{\alpha }t\\ & = & De^{-\delta t}\;t^{1-\alpha }\;E_{1,\;2-\alpha }^{2}\left(\delta t\right)-D\delta^{\alpha }t\;, \end{array}$$
that gives the short- and long-time behavior
(15)$$\langle x^{2}\left(t\right)\rangle =\left\{\begin{array}{cc} Dt^{1-\alpha }/\Gamma \left(2-\alpha \right) & \mathrm{if}\;t\to 0\;,\\ D\alpha \delta^{\alpha -1} & \mathrm{if}\;t\to \infty \;. \end{array}\right.$$
The interrelation in the conjugate pair between each other is quite non-trivial. If for the Laplace exponent $\Psi_{\mathrm{temp}}\left(s\right)$ the pure subduffusion evolves to normal diffusion in time, then for the conjugate case $s/\Psi_{\mathrm{temp}}\left(s\right)$ the subdiffusion transforms into diffusion-limited aggregation. Figure 1b just presents the evolution. It has a simple explanation. As the normal diffusion is characterized by the Laplace exponent Ψ(s)=s, its conjugate partner has Φ(s)=s/Ψ(s)=1. This clearly implies the confinement. Using asymptotic behavior of the MSD, we determine the crossover time $t_{x}^{⋆}={\left[\Gamma \left(2-\alpha \right)\;\alpha \;\delta^{\alpha -1}\right]}^{1/(1-\alpha )}$ between the diffusive regimes. Consequently, the duality relation between infinitely divisible subordinators allows one to generate a new impact scenario of traps, in which diffusion behaves less anomalous at short time and extremely anomalous at long times.

Using numerical methods, the propagator under the conjugate Laplace exponent $s/\Psi_{\mathrm{temp}}\left(s\right)$ is shown in Figure 2. The propagator has a cusp which is saved for t→∞. Recall that the tempered subdiffusion loses this feature at long times. If the axis *y* is logarithmic, as in Figure 2, the propagator of tempered subdiffusion goes to a parabola (see the panel a), whereas in the confined case it takes a triangular shape (panel b). This is not surprising because for t→∞ the propagator of diffusion motion with the conjugate Laplace exponent $s/\Psi_{\mathrm{temp}}\left(s\right)$ can be found analytically, and its form corresponds to the well-known Laplace (or double exponential) distribution [76,77], namely
(16)$$\lim_{t\to \infty }p\left(x,t\right)=\frac{1}{\sqrt{2D\alpha \delta^{\alpha -1}}}\;\exp\left(-\frac{2|x|}{\sqrt{2D\alpha \delta^{\alpha -1}}}\right)$$
with a location parameter μ=0 (in general, it may be nonzero) and a scale parameter $\theta =\sqrt{\alpha \delta^{\alpha -1}D/2}>0$. Although the PDF of the Laplace distribution is reminiscent of the normal distribution, they are different: the normal distribution is expressed in terms of the squared difference from the mean whereas the Laplace distribution is expressed in terms of the absolute difference from the mean. Therefore, the Laplace distribution has fatter (more precisely, moderate) tails than the normal distribution (with thin tails always) [68]. To get the Laplace distribution as the average value of elementary Gaussians, the necessary (“superstatistical”) distribution of the diffusivities is exponential [78,79,80]. It should be noticed that the Brownian yet non-Gaussian diffusion is not the same considering in this paper. The stationary Laplace distribution of particles’ motion also takes place in compartmentalized media [81]. The inverse cumulative distribution function of the Laplace distribution is equal to $x_{c}=-\theta \ln\left(2-2q\right)$. The value $x_{c}$ is such that any observation from this distribution with the scale parameter θ falls in the range $\left[0\;x_{c}\right]$ with probability 0<q<1 [77]. This allows one to estimate borders of the confinement region, taking into account the values *D*, α and δ. The Laplace distribution as a confined case is characteristic for the Brownian motion as a parent process. If the parent process becomes infinitely divisible, the confined distribution will be other and presented in the next section.

Note that for α=1/2 the MSD of the tempered subdiffusion and its conjugate partner coincide with each other at short times. The point is that the Laplace exponent $s^{1/2}$ is the only one convertible into itself by the duality relation between conjugate pairs of Laplace exponents [82]. If α>1/2, then for the same values α the MSD of tempered subdiffusion less anomalous than the MSD of its conjugate partner at short times. For α<1/2 the opposite happens. Usually the duality relation accelerates the subdiffusion more anomalous (in the sense of α<1/2) and slows down too fast subdiffusion (with α>1/2). This is especially evident for multi-scale anomalous diffusion [82].

## 4. Confined Distributions for Infinitely Divisible Motion

Now we apply our approach for infinitely divisible motion as a parent process, whereas the subordinator has the Laplace exponent $s/[{\left(s+\delta \right)}^{\alpha }-\delta^{\alpha }]$ leading to a confined distribution for t→∞. Without loss of generality the one-dimensional case will be represented. Consider any infinitely divisible motion by using the characteristic function in the form
(17)$$\hat{h}\left(k,t\right)=\int_{-\infty }^{\infty }e^{ikx}\;h\left(x,t\right)\;dx=e^{-D^{*}t\;\Xi \left(|k|\right)/2}\;,$$
where $D^{*}$ is a generalized diffusive constant. In the case of β-stable Lévy motion the characteristic exponent Ξ(|k|) is equal to ${\left|k\right|}^{\beta }$ with β∈ (0, 2). There are also other well-known examples of the characteristic exponent: (i) ${\left(|k\right|}^{2}+m^{\beta /2}{)}^{2/\beta }-m$, β∈ (0, 2); (ii) $\log(1+|k{|}^{\beta })$, β∈ (0, 2]; (iii) ${b|k|}^{2}+{\left|k\right|}^{\beta }$; (iv) $\log((1+{\left|k\right|}^{2})+\sqrt{{\left(1+|k\right|}^{2}{)}^{2}-1})$ and so on [69].

The next development is to consider the subordination of such parent processes. For this purpose we use the same subordinator led to the Laplace distribution above from Brownian motion. Based on the simple forms of $\hat{h}\left(k,\tau \right)$ and $\tilde{f}\left(\tau ,s\right)$, the solution (propagator) of a subordinated infinitely divisible motion is convenient to write as the Laplace–Fourier transform, taking the form
(18)$$\tilde{\hat{p}}\left(k,s\right)=\frac{1}{s}\frac{s/({\left(s+\delta \right)}^{\alpha }-\delta^{\alpha })}{[D^{*}\Xi \left(|k|\right)/2+s/\left({\left(s+\delta \right)}^{\alpha }-\delta^{\alpha }\right)]}\;.$$
As $\lim_{s\to 0}s/\left({\left(s+\delta \right)}^{\alpha }-\delta^{\alpha }\right)=\delta^{1-\alpha }/\alpha$, using the final value theorem ($\lim_{t->\infty }\hat{p}\left(k,t\right)=\lim_{s->0}\tilde{\hat{p}}\left(k,s\right)$), the confined characteristic function is written as
(19)$$\hat{p}\left(k,\infty \right)=\frac{1}{1+D^{*}\alpha \delta^{\alpha -1}\;\Xi \left(|k|\right)/2}\;.$$
For the ordinary Brownian motion it is not difficult to check this result by the inverse Fourier transform clearly, as this is the tabulated integral [71]. Other forms of the characteristic exponent Ξ(|k|) do not lead to so simple analytical expressions, but then they can be evaluated numerically. Note that 1/[1+AΞ(|k|)] with $A=D^{*}\alpha \delta^{\alpha -1}/2>0$ is an even function, and thus its Fourier transform is equivalent to the cosine transform.

Taking the β-stable Lévy motion with the characteristic exponent $\Xi \left(|k|\right)={\left|k\right|}^{\beta }$ under β∈ (0, 2), the confined characteristic function ${\hat{p}}_{\beta }\left(k,\infty \right)$ manifests the characteristic function of the symmetric Linnik distribution [83] (or the β-Laplace distribution, following Pillai [84]), namely
(20)$$\begin{array}{ccc} p_{\beta }\left(x,\infty \right) & = & \frac{1}{\pi }\int_{0}^{\infty }\frac{\cos(k|x|)}{1+Ak^{\beta }}\;dk\\ & = & \frac{1}{\pi }\int_{0}^{\infty }\frac{\sin(z^{1/\beta }\;|x|)}{{\left(1+Az\right)}^{2}}\;dz\;. \end{array}$$
The last expression was obtained from the integration by parts and has a better convergence in numerical integration. Examples are shown in Figure 3. The symmetric Linnik distribution attracted considerable attention from researchers [85,86,87,88]. Generally, the PDF is unimodal [89], geometrically stable [90] and can be expressed in terms of Meijer’s G-function [91]. Moreover, the peak of the density is finite for 1<β≤2 (see Figure 3a), it becomes infinite for 0<β≤1 (shown in Figure 3b) [92]. Based on the tabular integral of [93], its value yields
(21)$$p_{\beta }\left(0,\infty \right)=\frac{1}{\pi }\int_{0}^{\infty }\frac{dk}{1+Ak^{\beta }}=\frac{1}{\beta A^{1/\beta }\sin\left(\pi /\beta \right)}\;.$$
A series expansion for small *x* is written as
(22)$$p_{\beta }\left(x,\infty \right)=\frac{1}{\beta }\;\sum_{n=0}^{\infty }\frac{{\left(-1\right)}^{n}}{(2n)!}\frac{x^{2n}}{A^{(1+2n)/\beta }}\;\frac{1}{\sin[\pi (1+2n)/\beta ]}\;.$$
If x=0, only the n=0 term is saved, and one obtains the previous expression. According to [94], the asymptotic expansion for large *x* reads
(23)$$p_{\beta }\left(x,\infty \right)=\frac{1}{\pi }\;\sum_{n=1}^{\infty }{\left(-1\right)}^{n+1}\;\frac{\Gamma \left(1+n\beta \right)\;A^{n}}{{\left|x\right|}^{1+\beta n}}\;\sin\left(\pi \beta n/2\right)\;.$$
Consequently, the leading term of this expansion becomes
(24)$$p_{\beta }\left(x,\infty \right)∼\frac{\Gamma (1+\beta )\;\sin(\pi \beta /2)}{\pi }\;\frac{A}{{\left|x\right|}^{1+\beta }}\;.$$
There are some specific examples representable by tabular integrals [93] that will be considered elsewhere.

Since Ξ(|k|) is a Bernstein function (or otherwise the function having a complete monotone derivative), the characteristic function 1/[1+AΞ(|k|)] is typical for a geometrically infinitely divisible PDF [95]. In any case the PDF form is symmetric and unimodal. In dependence of Ξ(|k|) it has a finite or infinite maximum. This is because the integral $\int_{0}^{\infty }dk/\left[1+A\Xi \left(k\right)\right]$ has a single improper point, namely k→∞, where the integral is convergent or divergent.

We can formulate the following **Confinement Principle:** Any subordinated infinitely divisible motion, in which the subordinator is characterized by the Laplace exponent conjugate to a tempered α-stable process, has a confined probability distribution. By the infinitely divisible motion we mean a wide class of infinitely divisible processes, including Brownian motion (as a marginal case), Lévy stable motion (Lévy flight) and many other processes with jumps. It is important that each case of characteristic exponents in such an infinitely divisible motion determines its confined probability distribution. For the pure Brownian motion this is the Laplace distribution whereas for the Lévy flights its generalization is the Linnik distribution. This procedure covers a class of geometrically infinitely divisible distributions as a confined case of the infinitely divisible motion subordinated by a special subordinator responsible for the confinement.

## 5. Conditionally Non-Exponential Decay of Relaxation

Our comparative analysis of tempered and confined diffusion may be pretty simple extended to relaxation processes with non-exponential decay. As is well known [96,97], the manifestations of many-body effects in anomalous dynamics of relaxing systems, independent of the physical and chemical structures of their interacting entities, are successfully described by stochastic tools. Then the relaxation function of non-exponential relaxation is written as
(25)$$\phi_{\Psi }\left(t\right)=\int_{0}^{\infty }e^{-b\tau }\;f_{\Psi }\left(\tau ,t\right)\;d\tau \;,$$
where *b* is a constant, and $f_{\Psi }\left(\tau ,t\right)$ is the PDF of an inverse subordinator $S_{\Psi }\left(t\right)=\inf\left\{\tau >0\;:\;T_{\Psi }\left(\tau \right)>t\right\}$. The Laplace image $\phi_{\Psi }\left(t\right)$ takes the simple form
(26)$${\tilde{f}}_{\Psi }\left(\tau ,s\right)=\frac{\Psi (s)}{s}\;e^{-\tau \Psi (s)}\;.$$
Then the Laplace transform of $\phi_{\Psi }\left(t\right)$ in time yields
(27)$${\tilde{\phi }}_{\Psi }\left(s\right)=\frac{1}{s}\frac{\Psi (s)}{\Psi (s)+b}\;.$$
Similar conversions that we omit can be done for s/Ψ(s).

Based on the Laplace exponent of tempered diffusion $\Psi_{\mathrm{temp}}\left(s\right)={\left(s+\delta \right)}^{\alpha }-\delta^{\alpha }$ and its conjugate partner $s/\Psi_{\mathrm{temp}}\left(s\right)$, it is not difficult to obtain their relaxation functions numerically. They are presented in Figure 4.

Using the above relationship, we have found asymptotic behavior of the functions. They read
(28)$$\begin{array}{ccc} \lim_{t\to 0}\left(1-\phi_{\mathrm{temp}}\left(t\right)\right) & = & bt^{\alpha }/\Gamma \left(\alpha +1\right)\;,\\ \lim_{t\to 0}\left(1-\phi_{\mathrm{conf}}\left(t\right)\right) & = & bt^{1-\alpha }/\Gamma \left(2-\alpha \right) \end{array}$$
at short times and accordingly
(29)$$\begin{array}{ccc} \lim_{t\to \infty }\phi_{\mathrm{temp}}\left(t\right) & = & \lim_{t\to \infty }e^{-bt\delta^{1-\alpha }/\alpha }=0\;,\\ \lim_{t\to \infty }\phi_{\mathrm{conf}}\left(t\right) & = & \frac{1}{1+\alpha \delta^{\alpha -1}b}=\mathrm{const} \end{array}$$
at long times. Note that both types of relaxation start with 1 as a power function in time. However, tempered relaxation tends to zero exponentially, whereas the confined relaxation does not reach zero at all. From the physical point of view this latter model can be interpreted in the following way. Dipoles ordered by the external field do not fall into disorder with probability 1 after removing the field as *t* tends to infinity. Therefore, we believe that this model demonstrates a conditionally non-exponential decay. In this context it should be mentioned that the conditionally exponential decay model is a key for the concept of clusters and their dynamics to an imperfectly ordered state, used for the explanation of relaxation in dielectric materials [98,99,100]. During the relaxation process the strongly coupled local (intracluster) motions are expected to be generated first and then followed by the weakly coupled (intercluster) motions which produce the partial long-range structure. Each of these motions, those leading to the local structure order and those leading to the cluster ordering in general, has its own perceptible contribution to the observed features such as the relaxation function in time domain and the susceptibility in frequency domain.

## 6. G-Proteins vs. a2AR Receptors from the Analysis of the SPT Data

As an example for detecting the Laplace confinement in experimental data, we present our analysis of random trajectories obtained from a recent SPT study on G protein-coupled receptors, namely the motion and interaction of individual receptors and G proteins on the surface of living cells [44]. Two types of particles and only the basal case (without drug stimulation) data were studied: G-protein coupled receptors (further we will call them simply receptors) and the G proteins with which the receptors interact. The sample data consisted of 20,000 trajectories of 30 sets for G proteins and more 35,000 trajectories of 30 sets for receptors which randomly walk along *x* and *y* coordinates.

The first aim of the data study is to classify the dynamics for both types of particles. It is based on the standardized maximal distance $T_{n}$ of random processes from its starting point [101]. This approach is quite justified. Really, if the motion is driven by the fractional Brownian motion, then the best among the available methods is the one based on the *p-VAR* test, especially for longer trajectories. But, if the particle dynamics can be described by the Ornstein–Uhlenbeck or diffusive Brownian motion process, then the method yielding the smallest errors is based on the *MAX* test [102,103,104,105]. Following the procedure, we use the statistical test:$H_{0}$—an observed trajectory $X=\left\{X_{1},X_{2},\cdots ,X_{n}\right\}$ comes from Brownian motion,$H_{1}$—the trajectory looks like a confined or directed diffusion.

Then $T_{n}$ is estimated with respect to the quantiles of order α/2 and 1−α/2 (for example, α = 5%) for different trajectory lengths *n*. The decision rule is as follows: $T_{n}<q_{n}\left(\alpha /2\right)$ means a confined motion, whereas $T_{n}>q_{n}\left(1-\alpha /2\right)$ is superdiffusion (or directed motion). If $q_{n}\left(\alpha /2\right)<T_{n}<q_{n}\left(1-\alpha /2\right)$, then $X=\left\{X_{1},X_{2},\cdots ,X_{n}\right\}$ is Brownian motion. Consequently, this permits us to classify the trajectories available for processing. The results are shown in Figure 5. As seen from this figure, the most of trajectories is Brownian motion: 69% for G proteins and 78% for receptors. The contribution of superdiffusion is the smallest, 2%. The rest corresponds to confined motion. This part is especially interesting to us. Next, we are going to estimate the statistics of such trajectories. It is assumed that the confined random walks can occur in two cases. The first of them is classical, the Ornstein–Uhlenbeck model. It gives the normal distribution for t→∞. The second case leads to the Laplace confinement for t→∞, considered above. Possible transitions of the particles’ diffusion type within single trajectories are noted and investigated. For example, in [104] it has been proposed a statistical procedure for detecting transitions of the MSD exponent value within a single trajectory.

Discriminating the statistics of G-protein and receptor confined trajectories between the normal and Laplace distribution functions, we apply the logarithm of the ratio of their maximized likelihoods [106]. The approach leads to the calculations of means, medians, sample variances and averages of the absolute difference between data values and the median. This statistical test gives the ratio Q>0 for the normal distribution, otherwise the Laplace distribution is preferred. After applying the second statistical test, its results together with the first test results are also presented in Figure 5. This shows that for G proteins the confined trajectories obey equally the normal and Laplace distributions, whereas for receptors the normal distribution is approximately twice as common as the Laplace distribution. But it should be pointed out also that the share of confined trajectories with normal statistics remains unchanged for both G proteins and receptors. Judging by the contribution of Brownian motion in all the sets of trajectories, the difference between the percentage ratio of confined trajectories with the normal and Laplace distributions for G proteins and receptors can indicate greater mobility of receptors over G proteins. It should be mentioned that Laplace distributions were detected in the complex diffusive behavior of RNA-protein particles [107].

The occurrence of the Laplace distribution for confined trajectories in the experimental data used by us seems to be natural. First, the most part of the trajectories is Brownian motion. What could be a parent process for subordination in this environment? Brownian motion is preferred. Why? Since we observe a following **Competition Principle**
*between parent processes: Brownian motion, Lévy motion or other infinitely divisible process even for any fixed subordinator conjugated one to tempered α-stable responsible for confinement*. If Brownian motion is parent, the confined distribution from our subordination approach can have only the Laplace form. In the above data sets any feature, for example, typical for Lévy motion, is not detected. If this was true, it would be a chance for the play of generalized Laplace distributions as a confined distribution. Another case is the Ornstein–Uhlenbeck process leading to the normal statistics in confined trajectories, it has the same (Brownian) roots too. Therefore, the presence of normal and Laplace distributions together into confined trajectories is quite logical and justified physically.

## 7. Discussion

We have revealed that the conjugate property of Bernstein functions connects the tempered stable subdiffusion with the diffusion-limited aggregation by an one-to-one mapping (in fact, a bijection). If the pure subdiffusion is characterized by multiple trapping events with infinite mean sojourn time, and the power function exponent of MSD is constant in time, then a truncated power-law distribution of trapping times leads to tempered subdiffusion, in which diffusion is anomalous at short times and normal (contribution of traps seems to disappear) at long times [45]. The interpretation of anomalous diffusion tending to the confinement is that the trap impact has the opposite tendency, long waiting times in traps dominate more and more so that it becomes impossible to leave such traps. This model, just like the tempered one, is applicable for the analysis of SPT. Its effects are present in confined random motions of G proteins and receptors in living cells. We have established that the confined distribution form depends on the PDF of the parent process under subordination. If the parent process is Brownian motion, the confined distribution has only the Laplace form. If the Lévy motion is directed, the confined distribution takes the Linnik case. If the support of the parent process is changed from (−∞,∞) to (0,∞), as a confined limit, the Mittag–Leffler distribution arises. All this manifests that the presented method has ample opportunities for the study of confined random walks in complex systems. Concerning to relaxation phenomena, complete disorder (e. g. in the form of charge neutralization in dielectrics) does not occur in the relaxing system with confinement features. This concept can be used for developing new cluster models of non-exponential relaxation. It will be considered in more detail elsewhere. Our new methodology is generally valid in a wide class of problems of transport in random media that include live cells, relaxation in heterogeneous substances, and jump-diffusion.

## Figures and Tables

**Figure 1 entropy-22-01317-f001:**
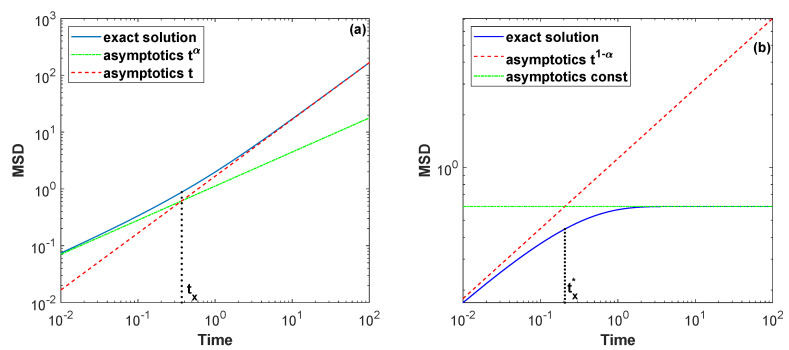
(Color online) Mean squared displacement of tempered subdiffusion (**a**) and its conjugate partner (**b**) with α=0.6 and δ=1 (for D=1). The dashed red and dash-dot green lines show asymptotic behavior of the values. If the panel (**a**) indicates a transition of the subdifussion into normal diffusion at long times, whereas the panel (**b**) shows the emergence of diffusion-limited aggregation.

**Figure 2 entropy-22-01317-f002:**
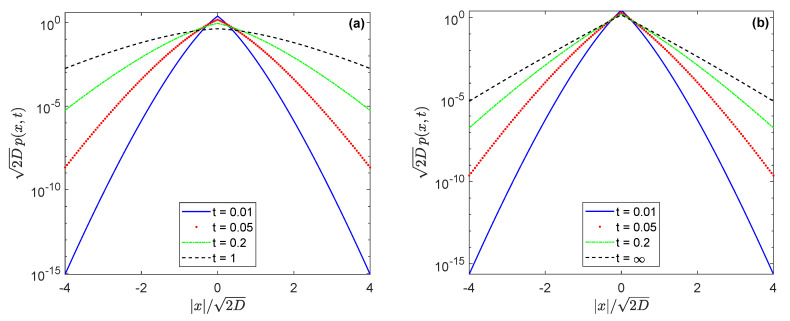
(Color online) Propagator p(x,t) for the tempered subdiffusion (**a**) and its conjugate partner (**b**), tending to the confinement, with a constant potential, α=0.5 and δ=1, drawn for consecutive dimensionless instances of time. Starting with the Dirac delta-function and passing to the subdiffusive PDF, for t→∞ the value p(x,t) becomes the normal distribution, shown by black dotted line on the panel (**a**), and the Laplace distribution (black dotted line) on the panel (**b**).

**Figure 3 entropy-22-01317-f003:**
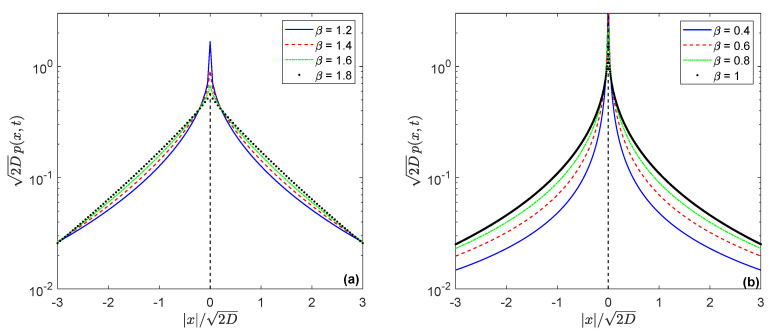
(Color online) Propagators p(x,t) from the parent processes, having the β-stable Lévy distribution: (**a**) 1<β<2; (**b**) 0<β≤1; under the subordinator, conjugate to a tempered random process in the sense of Bernstein functions, for t→∞. The value $A=D^{*}\alpha \delta^{\alpha -1}/2$ is taken equal to 1.

**Figure 4 entropy-22-01317-f004:**
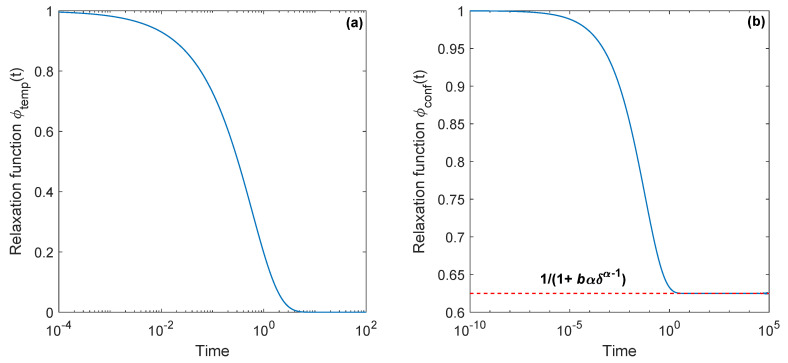
(Color online) Relaxation functions, caused by the inverse tempered subordinator (**a**) and its conjugate partner (**b**) respectively, with α=0.6, δ=1 and b=1. The first represents the tempered relaxation, and the second is confined. The dashed red line shows a conditionally non-exponential decay due to the confinement effect ($\lim_{t\to \infty }\phi_{\mathrm{conf}}\left(t\right)=\mathrm{const}\neq 0$).

**Figure 5 entropy-22-01317-f005:**
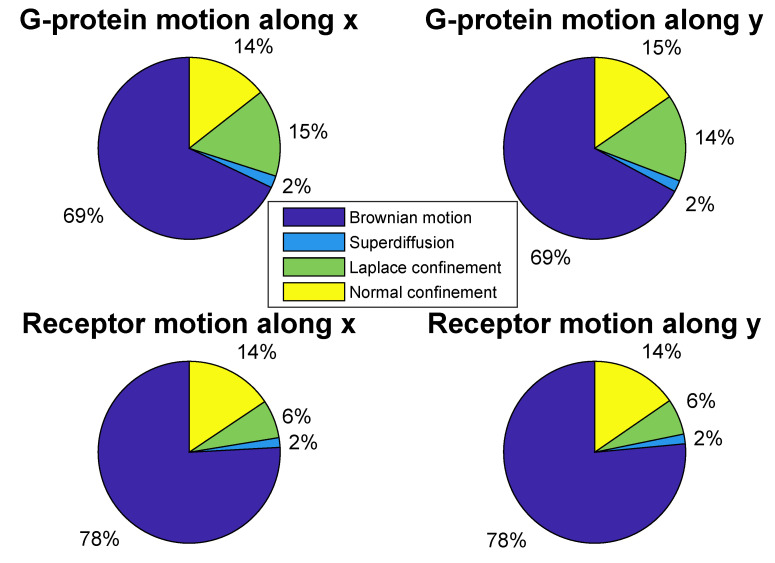
(Color online) Analysis of the experimental data as applied to G-protein and receptor random-walk trajectories along the coordinates *x* and *y* with the cutoff length of trajectories more and equal to 50.

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
