# Peer review of "Look at Tempered Subdiffusion in a Conjugate Map: Desire for the Confinement"

_entropy, 2020, doi:10.3390/e22111317_

Round 1
Reviewer 1 Report
It was a pleasure to read this manuscript by two well known experts in the
field of stochastic processes. They consider how Laplace distributions may
arise from the concept of Bernstein functions. I find the study timely and
interesting, both from a mathematical and a physical perspective. After a
minor revision this manuscript should be published in Entropy.
(1) The authors mention the trap-transition to motivate the emergence of a
full stalling (plateau) of the MSD, but maybe this point could be expanded
somewhat from a physical point of view. In the G protein example can this
effect be discussed using real data? If available I would appreciate a lot
a more quantitative discussion, on top of the pie diagrams in figure 5.
(2) The authors should connect this stalling behaviour with the discussion
of Meerschaert's tempered FBM in New J Phys 20, 103027 (2018).
(3) In addition to the mentioned references for Laplace distributions, the
authors should mention the interesting work reported in Phys Rev Lett 124,
060603 (2020), as well as arXiv:1909.11395.
(4) Finally, when Laplace distributions and/or crossover behaviour of ano-
malous diffusion are discussed in the text, the authors could include some
experimental studies, such as Biophys J 112, 532 (2017). Experiments plus
modelling are discussed in Phys Rev Lett 125, 058101 (2020) and crossovers
for different cases are analysed in New J Phys 22, 083041 (2020).
Author Response
We thank the Referee very much for his/her constructive evaluation of our work. We have followed all of the Referee’s suggestions point by point.
(1) The authors mention the trap-transition to motivate the > emergence of a full stalling (plateau) of the MSD, but maybe this point could be expanded somewhat from a physical point of view. In the G protein example can this effect be discussed using real data? If available I would appreciate a lot a more quantitative discussion, on top of the pie diagrams in figure 5.
This work is in progress. Really, in the G protein example the drugs change the contributions of Brownian motion, Laplace confinement and normal confinement. It is clear, if the confinement in life cells dominates, then this is bad for such a cell. Brownian motion without confinement is not good too. What should be the ratio between them for cells to be OK is an interesting question. We work on the problem and cannot open all the details now.
(2) The authors should connect this stalling behaviour with the discussion of Meerschaert's tempered FBM in New J Phys 20, 103027 (2018).
The paper New J Phys 20, 103027 (2018) [58] is mentioned in our paper. From our point of view the discussion of Meerschaert's tempered FBM is something like a step aside. We have decided not to include it for not overloading our work.
(3) In addition to the mentioned references for Laplace distributions, the authors should mention the interesting work reported in Phys Rev Lett 124, 060603 (2020), as well as arXiv:1909.11395.
Phys Rev Lett 124, 060603 (2020) is in our references [41]. The reference to arXiv:1909.11395 is added below Eq. (16).
(4) Finally, when Laplace distributions and/or crossover behaviour of anomalous diffusion are > discussed in the text, the authors could include some experimental studies, such as Biophys J 112, 532 (2017). Experiments plus modelling are discussed in Phys Rev Lett 125, 058101 (2020) and crossovers for different cases are analysed in New J Phys 22, 083041 (2020).
Biophys J 112, 532 (2017) was included in the end of Section 6. As for Phys Rev Lett 125, 058101 (2020) and New J Phys 22, 083041 (2020), they are outside our work.
Reviewer 2 Report
This is a good professional paper employing subordination
concept for the description of different regimes of anomalous
diffusion. Such concept is widely used and well-known, however
the novelty of the manuscript lies in the invention of the
confined regimes for the specific forms of the subordinators.
In this context the relaxation leading to the confined state
looks like an antipode of the known case with tempered
subordinator. Interestingly, the distribution in the confined
regime is not a Gaussian one, contrary to e.g., Ornstein-Uhlenbeck confinement, but rather double exponential. The authors not only make a beautiful mathematical analysis of the phenomenon, but also provide a data analysis of the single particle tracking experiment demonstrating the existence of such Laplace confinement. I am a bit sceptical about the authors' conclusions regarding the experiment,
nevertheless I support the inclusion of this section into the manuscript as a first step that may encourage experimentalists for future studies and thus increase the visibility of the paper. Summarizing the first part of my report, in my opinion the paper with no doubts has a potential
to be published in the Entropy. However, I have several concerns which the authors should address before the final decision could be made.
1. The Laplace regime found in the manuscript should not be mixed with the Laplace regime of the Brownian yet
non-Gaussian diffusion which is widely observed in soft matter
and biological systems. This should be clearly stated.
2. The physical interpretation of the results looks doubtful.
In the Discussion section the authors talk about traps
disappearing or growing in time. Does it mean non-stationarity
of the environment ? If so, I can not believe that the authors'
methodology "is generally valid in a wide class of problems
of transport in random media" as they claim at the end
of the paper, since most of random media they refer to are in
equilibrium or at least close to equilibrium. While discussing
this issue the authors should comment on the difference with the physical interpretation of retarded and accelerated diffusion
governed by distributed order diffusion equations. Such interpretation was given in Sandev et al., PRE 92, 042117 (2015), and refer to the mix of traps with different waiting time distributions and pooling of continuous time random walks, respectively.
3. While I agree with the "Confinement Principle" introduced by
the authors, I can not accept the "Competition Principle".
For me it looks like a combination of meaningless phrases. I kindly ask the authors either to remove it at all or to rephrase and extend the discussion. In particular, while talking about representation of heterogeneities of the environment, the authors should refer to such representations in the anomalous diffusion context as heterogeneous diffusion process, see Cherstvy et al., NJP 15, 083039 (2013), and Brownian yet non-Gaussian diffusion in heterogeneous media, see Postnikov et al., NJP 22, 063046 (2020).
4. When referring to retarding and accelerating subdiffusion,
would be a good manner to cite not only the own paper of 2020, but also much older papers where these types of diffusion have been introduced for the first time.
Author Response
Many thanks the Referee for the detailed evaluation of our work and useful critical comments.
1. The Laplace regime found in the manuscript should not be mixed with the Laplace regime of the Brownian yet non-Gaussian diffusion which is widely observed in soft matter and biological systems. This should be clearly stated.
We agree. After Eq. (16) we state that the Brownian yet non-Gaussian diffusion is not considered in our paper.
2. The physical interpretation of the results looks doubtful. In the Discussion section the authors talk about traps disappearing or growing in time. Does it mean non-stationarity of the environment ? If so, I can not believe that the authors' methodology "is generally valid in a wide class of problems of transport in random media" as they claim at the end of the paper, since most of random media they refer to are in equilibrium or at least close to equilibrium. While discussing this issue the authors should comment on the difference with the physical interpretation of retarded and accelerated diffusion governed by distributed order diffusion equations. Such interpretation was given in Sandev et al., PRE 92, 042117 (2015), and refer to the mix of traps with different waiting time distributions and pooling of continuous time random walks, respectively.
This does not mean non-stationarity of the environment. The probabilistic properties of waiting times of traps are that the MSD exponent changes in time, and it looks like the evolution of trap properties in time. The part of Discussion was clarified. We write the following:
“If the pure subdiffusion is characterized by multiple trapping events with infinite mean sojourn time, and the power function exponent of MSD is constant in time, then a truncated power-law distribution of trapping times leads to tempered subdiffusion, in which diffusion is anomalous at short times and normal (contribution of traps seems to disappear) at long times [45]. The interpretation of anomalous diffusion tending to the confinement is that the trap impact has the opposite tendency, long waiting times in traps dominate more and more so that it becomes impossible to leave such traps.”
The interpretation given in Sandev et al., PRE 92, 042117 (2015) is controversial because this cannot be a physical mix of traps, and “a combination of two kinds of trapping landscapes” is only a play on words. In fact, there are traps with the PDF in the form of a sum rather than “the mix of traps”.
3. While I agree with the "Confinement Principle" introduced by the authors, I can not accept the "Competition Principle". For me it looks like a combination of meaningless phrases. I kindly ask the authors either to remove it at all or to rephrase and extend the discussion. In particular, while talking about representation of heterogeneities of the environment, the authors should refer to such representations in the anomalous diffusion context as >heterogeneous diffusion process, see Cherstvy et al., NJP 15, 083039 (2013), and Brownian yet non-Gaussian diffusion in heterogeneous media, see Postnikov et al., NJP 22, 063046 (2020).
We do not talk about a heterogeneous medium with space dependent diffusivity. Therefore, we do not see the need for references to Cherstvy et al., NJP 15, 083039 (2013) and Postnikov et al., NJP 22, 063046 (2020). In our case the heterogeneities are due to specific traps. Heterogeneous media leading to Brownian yet non-Gaussian diffusion are not the same describing in our paper (see 1. above). From our point of view "Confinement Principle" represents different structure of environment becoming heterogeneous because of subordination.
We simplified the Confinement Principle in the following way:
The occurrence of the Laplace distribution for confined trajectories in the experimental data used by us seems to be natural. First, the most part of the trajectories is described by Brownian motion. What could be a parent process for subordination in this environment? Brownian motion is preferred. Why ? Since we observe here a following “Competition Principle” between parent processes: Brownian motion, Lévy motion or other infinitely divisible process even for any fixed subordinator conjugated one to a tempered a-stable responsible for confinement. If Brownian motion is parent, the confined distribution from our subordination approach can have only the Laplace form. In the above data sets any feature, for example, typical for Lévy motion, is not detected. If this was true, it would be a chance for the play of generalized Laplace distributions as a confined distribution. Another case is the Ornstein-Uhlenbeck process leading to the normal statistics in confined trajectories, it has the same (Brownian) roots too. Therefore, the presence of normal and Laplace distributions together into confined trajectories is quite logical and justified physically.
4. When referring to retarding and accelerating subdiffusion, would be a good manner to cite not only the own paper of 2020, but also much older papers where these types of diffusion have been introduced for the first time.
The second sentence of Discussion was removed.
Reviewer 3 Report
The Authors consider an interesting application of the tempered processes in description of the experimental data with confined random trajectories of G proteins and receptors in living cells. The paper is well written and of current interest to the journal readership. Therefore, the manuscript is worth to be published in the journal Entropy.
In what follows I include some comments on the manuscript.
- The Authors state that the Laplace exponent should be a Bernstein function. Can you say something about the non-negativity of the PDF p(x,t)? What are the restrictions on the memory kernel M(t) in order p(x,t) be non-negative?
- Similar analysis of two different forms of the Fokker-Planck equation where the memory kernels are conjugate pairs as in the current manuscript was done, for example, in [Chaos, Solitons & Fractals 102, 210-217 (2017)] and [Fractional Calculus and Applied Analysis 21 (1), 10-28 (2018)]. Same behavior for the MSDs in case of exponentially truncated power-law memory kernel was obtained: for one of the cases, the MSD approaches normal diffusion while for the case of conjugate Laplace exponent the MSD approaches saturation [Chaos, Solitons & Fractals 102, 210-217 (2017)]. So the Authors may refer to these papers.
- Eq. (19): the Authors may mention that the stationary distribution in the long time limit can be obtained from Eq. (18) by using the final value theorem, \lim_{t->\infty}p(k,t)=\lim_{s->0}sp(k,s)
- The Authors may mention that such characteristic crossover form subdiffusion to normal diffusion has been observed in lipid bilayer systems [Phys. Rev. Lett.109, 188103 (2012], [Phys. Rev. E79, 011907 (2009)], [New Journal of Physics 20, 103027 (2018)]
Author Response
We thank the Referee for his/her useful suggestions to our work.
̶ The Authors state that the Laplace exponent should be a Bernstein function. Can you say something about the non-negativity of the PDF p(x,t)? What are the restrictions on the memory kernel M(t) in order p(x,t) be non-negative?
This is very simple. The exponential function $e^{-at}$ is completely monotone (CM). The exponential function of a Bernstein function is CM again. The inverse Laplace transform of a CM function is non-negative and viceversa. The memory kernel is CM, i.e. it is a non-negative with infinitely many derivatives that alternate in sign.
̶ Similar analysis of two different forms of the Fokker-Planck equation where the memory kernels are conjugate pairs as in the current manuscript was done, for example, in [Chaos, Solitons & Fractals 102, 210-217 (2017)] and [Fractional Calculus and Applied Analysis 21 (1), 10-28 (2018)]. Same behavior for the MSDs in case of exponentially truncated power-law memory kernel was obtained: for one of the cases, the MSD approaches normal diffusion while for the case of conjugate Laplace exponent the MSD approaches saturation [Chaos, Solitons & Fractals 102, 210-217 (2017)]. So the Authors may refer to these papers.
Referred in the end of Section 2.
̶ Eq. (19): the Authors may mention that the stationary distribution in the long time limit can be obtained from Eq. (18) by using the final value theorem, \lim_{t->\infty}p(k,t)=\lim_{s->0}sp(k,s)
Added after Eq.(18).
̶ The Authors may mention that such characteristic crossover form subdiffusion to normal diffusion has been observed in lipid bilayer systems [Phys. Rev. Lett.109, 188103 (2012], [Phys. Rev. E79, 011907 (2009)], [New Journal of Physics 20, 103027 (2018)]
Added in the end of Introduction.